# High-Performance, Easy-to-Fabricate, Nanocomposite Heater for Life Sciences and Biomedical Applications

**DOI:** 10.3390/polym16081164

**Published:** 2024-04-20

**Authors:** Yudan Whulanza, Husein Ammar, Deni Haryadi, Azizah Intan Pangesty, Widoretno Widoretno, Didik Tulus Subekti, Jérôme Charmet

**Affiliations:** 1Department of Mechanical Engineering, Faculty of Engineering, Universitas Indonesia, Depok 16424, Indonesia; 2Research Center for Biomedical Engineering, Universitas Indonesia, Depok 16424, Indonesia; 3Department of Mechanical Engineering, Gunadarma University, Depok 16424, Indonesia; 4Department of Metallurgical and Materials Engineering, Faculty of Engineering, Universitas Indonesia, Depok 16424, Indonesia; 5Research Organization for Health, National Research and Innovation Agency, Central Jakarta 10340, Indonesia; wido009@brin.go.id (W.W.); didi028@brin.go.id (D.T.S.); 6School of Engineering HE-Arc Ingénierie, HES-SO University of Applied Sciences Western Switzerland, 2000 Neuchâtel, Switzerland; 7Faculty of Medicine, University of Bern, 3010 Bern, Switzerland

**Keywords:** nanocomposites, microheaters, screen printing, LAMP

## Abstract

Microheaters are used in several applications, including medical diagnostics, synthesis, environmental monitoring, and actuation. Conventional microheaters rely on thin-film electrodes microfabricated in a clean-room environment. However, low-cost alternatives based on conductive paste electrodes fabricated using printing techniques have started to emerge over the years. Here, we report a surprising effect that leads to significant electrode performance improvement as confirmed by the thorough characterization of bulk, processed, and conditioned samples. Mixing silver ink and PVA results in the solubilization of performance-hindering organic compounds. These compounds evaporate during heating cycles. The new electrodes, which reach a temperature of 80 °C within 5 min using a current of 7.0 A, display an overall 42% and 35% improvement in the mechanical (hardness) and electrical (resistivity) properties compared to pristine silver ink electrodes. To validate our results, we use the composite heater to amplify and detect parasite DNA from Trypanosoma brucei, associated with African sleeping sickness. Our LAMP test compares well with commercially available systems, confirming the excellent performance of our nanocomposite heaters. Since their fabrication relies on well-established techniques, we anticipate they will find use in a range of applications.

## 1. Introduction

Whether to control temperature-dependent processes [1,2,3,4,5], drive actuators [6,7,8], or as part of sensing mechanisms [4,9,10,11,12], microheaters have found applications in the chemical, food processing, automotive, aeronautic, and biomedical and pharmaceutic industries, to cite a few (see, e.g., ref. [11,13,14,15,16,17,18]).

In addition to the obvious miniaturization aspects, microheaters have distinct advantages over their macroscale counterparts. An important feature is their superior operation efficiency when combined with resistive (or Joule) heating. The heating power *p* = RI^2^, where I is the current passing through a conductor of resistance R = r∙L/A, can be written as follows:(1)$$P=\;\frac{\rho \cdot L\cdot I^{2}}{A}$$

Equation (1) highlights the heating power’s dependency on the material’s property (resistivity ρ) and geometry of the heating element (length L and cross section A). At the microscale, the heating is due to the loss of kinetic energy by charged particles (electrons in our case) travelling through the medium [19]. Therefore, for a given material, microheaters, manufactured using conventional microfabrication techniques, which typically enable the fabrication of long (centimeter scale) conductors with small (microscale) cross-section, are ideally suited to maximize heating power.

Another advantage of operating at the microscale is the increased surface-to-volume ratio that favors thermal transfer. We borrow examples from microfluidics where this feature is used to quickly heat up solutions, for example, in DNA amplification (e.g., polymerase chain reaction [20,21] or loop-mediated isothermal amplification [1,17]) or for cell culture in bioreactors [22,23] and also in electrophoretic separation where large electric fields can be applied without excessive heating due to improved thermal dissipation under flow [24,25]. The above observations highlight the excellent performances of Joule-based microheating, which enables fast, precise, and accurate control of temperatures with low power consumption, as confirmed in other studies [16,26].

In terms of dimensions, microscale cross-sections can be achieved through a combination of thin films deposition and structuring technique [3,7,27]. A notable example is lift-off (which relies on a combination of physical vapor deposition and photolithography) to create microheaters of cross sections, typically of a few µm^2^ (see, e.g., ref. [18]). In comparison, the smallest wires (gauge 40) have a cross-section of a few thousands µm^2^, which corresponds to a decrease in the heating power of the same factor (thousand-fold) for the same materials and length (if one ignores the contribution of size-dependent losses [16]). This emphasizes the importance of fabrication methods for the manufacturing of microheating elements. Yet, thin-film deposition methods and photolithography that allow for very small cross sections are usually limited to specialized laboratories [28]. This is mostly due to large investment and running costs. To overcome this limitation, low-cost alternatives have started to emerge.

More accessible methods to create low-cost microheaters include screen printing [16], doctor-blade techniques [29], 3D printing [14], or even inkjet printing [1]. However, compared to their thin films counterparts, the heating elements (electrodes) usually have a larger cross section, reaching dimensions that are more comparable to conventional wires. Therefore, they lose some of the microscale size related advantages highlighted above and, in particular, the heating power. 

To resolve this issue, we developed a composite ink made of PVA and silver ink. Our rationale was that the addition of a non-conducting matrix would result in a high resistivity ink capable of generating more heat for the same electrode dimensions and current (Equation (1)). The reason for choosing PVA as opposed to other polymer matrices resides in its non-toxicity and its solubility in water that make it an environmentally friendly solution. The fact that it is widely available, including in resource limited settings, also contributed to its selection. We also note that other silver-based composites have been studied elsewhere [30,31,32,33]. We fabricated microelectrodes from this nanocomposite using doctor-blade techniques with PVA concentrations up to 30% *w*/*w*. We optimized the electrodes fabrication and characterized their performance before and after heating cycles (Section 3.1 and Section 3.2). Interestingly, we noted that heating cycles lead to a stable improvement in the electrical and mechanical properties, regardless of the PVA concentration (Section 3.2). Improvements of 35% and 42% are noted for resistivity and hardness, respectively, for a nano-ink with 30% PVA compared to pristine silver ink samples. These surprising results prompted us to perform systematic studies on the composite materials in their unprocessed (Section 3.3) and processed conditions (Section 3.4). Our results show that the observed improvements are due to the removal of volatile organic compounds which evaporate during the heating cycles. These compounds found in the pristine silver ink dissolve when PVA is added. Their removal favors the affinity between PVA and the silver ink in the processed samples, which results in higher resistivity and hardness. 

To validate our results, we demonstrate that our composite microheater can be used to reliably detect the Trypanosoma brucei DNA using a LAMP test (Section 3.5). Due to the good stability of our heater, the results are comparable to those of commercially available systems in the same timeframe. Figure 1 shows an overview of the various geometries and characterization techniques applied herein.

## 2. Materials and Methods

*Preparation of silver–PVA ink.* Firstly, a 0.01 g/mL solution of polyvinyl alcohol (PVA from Sigma Aldrich, Darmstadt, Germany) was prepared by mixing a PVA pellet into distilled water. The mixture was then heated to 60 °C and stirred until complete dissolution of the pellet. Then, the solution was mixed with 1.5 g of silver ink (MJ Chemical, Jakarta, Indonesia) to obtain mixtures of PVA/silver with concentrations of 10%, 20%, and 30% *w*/*w*, referred to herein as 10% PVA, 20% PVA, and 30% PVA. Pristine silver ink is sometimes referred to as 0% PVA.

*Preparation of PVA/silver heating pad.* The silver ink was printed on acrylic substrate to create the heating electrodes using the doctor-blade method [29]. Circular test patterns with a 6 mm diameter were designed for hardness measurement. Rectangular strips (5 × 30 mm) were used for heating characterization (see Appendix A). These patterns were drawn using the Silhouette Studio v. 2.7.4 software and transferred to an adhesive vinyl film using an electronic craft cutter (Silhouette Cameo, Silhouette America, Inc., Lindon, UT, USA). The resolution was set to 300 DPI. Next, a patterned vinyl was attached to the acrylic chip. The doctor blade then applied the silver ink onto the substrate through the pattern while removing the excess ink. The ink was left to dry for 5 min before peeling off the vinyl. Finally, the ink was cured for 15 min at 120 °C on a hot plate.

*Geometrical Measurement.* A series of measurements were conducted to evaluate the dimension and morphology of the electrodes. The width of the electrodes was measured using a calibrated digital microscope Dinolite AM4113 (Anmo, New Taipei, Taiwan). The thickness and contour of electrode was observed using an Accretech Surfcom 2900SD3 from Seimitsu (Tokyo, Japan). Micrographs of the silver electrodes surface were acquired using Scanning Electron Microscope Inspect F50 (FEI Company, Hillsboro, OR, USA) with magnifications of 200× and 1000×.

*Electrical property measurement.* The resistivity of our material was determined by measuring the resistance of electrodes of known dimensions using a four-point probe method (see Appendix A for more details). The value of resistivity, defined as r = RA/L, plays an important role since it determines the temperature that can be dissipated due to the resistive (Joule) heating phenomenon.

*Mechanical property measurement.* Mechanical properties were evaluated using a universal tensile machine MCT series from A&D (Tokyo, Japan) with a maximum capacity of 500 N. The compression (mm) and load (N) were obtained at a crosshead speed of 1 mm/min and ended at 400 N of compression force.

*Heating performance measurement.* To generate heat through the Joule heating effect, a DC power supply (GW Instek (Taipei, Taiwan)) was connected to the electrodes. The test was carried out at a constant current. The resulting temperature change in the electrode was monitored using a four-channel data acquisition system. The temperature was monitored in real-time with NTC thermistor (TDK Electronics, Munich, Germany) connected to a data acquisition module from National Instrument NI-USB 6001 (Austin, TX, USA). An infrared (IR) camera (FLIR Thermovision A320) (Santa Barbara, CA, USA) was used to confirm the measurement. See Appendix A for a schematic of the set-up. Some measurements were performed by setting a voltage value instead of a current as explained in Appendix A.

*Porosity measurement*. The porosity of microstructure was measured using an image processing method using automatic global (histogram-derived) thresholding reported previously [34]. The protocol consists of five steps which are as follows: (1) pixel size unit conversion to metric units; (2) image segmentation; (3) morphological filter to split connected pores; (4) verification step; (5) extraction of statistics from the detected pores. ImageJ version 1.52j is the primary software used for this study, and analysis is performed on SEM images of 3D scaffolds. ImageJ is a public domain image-processing tool developed by the National Institutes of Health (ImageJ version 1.52j, 2018) and was used to perform analyses on the SEM micrographs.

*DNA Amplification*: A heating pad of 30% PVA–silver ink composite was deposited on a reaction chamber chip (Fluidic 584 series from Microfluidic Chipshop GmbH (Jena, Germany). The chip was then connected to a DC power supply GW Instek (Taipei, Taiwan) set to 4.5 A to reach 60 °C for 60 min. The reactor chambers have a volume of 20 µL. The sample consist of a PCR Mix 10 µL (KODC-PCR reagent, Konimex, Solo, Indonesia), Primer Forward 1 µL (WOAH 2022), and Primer reverse 1 µL (WOAH 2022). RNase-free water was 6 uL (KODC-PCR, Solo, Indonesia), and the DNA sample was 2 µL. Tripasonoma DNA was extracted from isolated Trypanosomes, purified by anion exchange chromatography using diethylaminomethyl. The sequence was extracted and validated using PCR by the Indonesian Research Center for Veterinary Science, Agricultural Research and Development Agency, Indonesian Ministry of Agriculture [35]. Buffer was used as negative control. The temperature was set to 60 °C for 60 min. A control amplification was performed on commercial thermocycler AllInOneCycler Bioneer (Daejeon, Republic of Korea) operated in the same conditions. 

*Gel electrophoresis*: After PCR amplification, the following amplification products were detected using agarose gel electrophoresis: 1 KB DNA Ladder IV catalog number DL006 (Geneaid, New Taipei, Taiwan) and agarose powder catalog number PC0701 (Vivantis Technologies, Shah Alam, Malaysia) and Florosafe DNA stain catalog number BIO-5170 (1st Base, Singapore, Singapore). For the staining of the agarose gel, loading dye catalog number HH 13,701 (Geneaid Biotech, New Taipei, Taiwan) was run at 80 V for 30 min in the MUPID-exU electrophoresis system (Advance, Tokyo, Japan). The resulting gel images were captured by a camera. Transilluminator UVstar 8 (Analytik Jena, Jena, Germany) was used to read the gels. FLIR thermal camera was used to confirm the heating process.

## 3. Results and Discussions

### 3.1. Fabrication of the Electrodes and Optimisation

Our electrodes are structured using doctor-blade processing [28,29]. Given the importance of electrode dimensions on Joule heating (Equation (1)) and since ink properties play a significant role on the resolution of structures obtained using doctor-blade techniques, we designed a circular test pattern to evaluate the resolution of the transfer process. Figure 2a shows the results for the silver 30% PVA nanocomposite. The transferred patterns show a good consistency with the original design despite the geometrical limitations inherent to colloidal systems, as reported elsewhere [36]. 

The influence of the PVA concentration on the transferred geometry is depicted in Figure 2b,c. An increase in PVA concentration results in a significantly thicker electrode and a slightly reduced diameter for the same doctor-blade processing conditions. The average diameter of the 30% PVA pattern is approximately 6% lower compared to that of pristine silver pad (0% PVA). Its thickness shows almost a two-fold increase. Each set of dimensions for both conditions (30% vs. 0% PVA) are significantly different, as indicated by the *p*-values of 0.007 and 0.008 for the diameter and thickness, respectively. The seemingly large standard deviations noted on the thickness values are in line with the typical values reported using doctor-blade techniques [37]. The increased interaction between the PVA and the Ag particles noted elsewhere [38] could explain the geometrical results, whereby increased cohesive forces increases layer thickness during processing and promotes lateral shrinkage upon drying. 

In parallel, we studied the effect of PVA on the specimen’s mechanical properties using Vickers microhardness measurements. Although a slight positive trend is observed, Figure 2d shows that the PVA concentration does not significantly alter the hardness compared to the pristine silver sample (0% PVA), as indicated by one-way ANOVA (F = 3.11, *p* = 0.067) and confirmed by Tukey’s test. We postulate that the large hardness deviation observed is caused by the heterogeneity of the nanocomposite (Figure 2a). 

Next, we set out to characterize the resistivity of the silver–PVA nanocomposite using the four-point probe technique, as shown in Appendix A. The results in Figure 2e show, as expected, that the resistivity increases with the addition of non-conducting PVA. It increases exponentially by approximately a factor of 2 for each 10% PVA added. A 7-fold increase to 0.33 mΩcm is noted for the 30% PVA sample compared to the pristine silver ink. The increase in resistivity is due to a reduction in the percolation path, due to a denser subphase that separates the conducting silver nanoparticles, and the Coulomb blockade effect, as demonstrated previously in similar composites [39].

### 3.2. Performances of the Nanocomposite Heater

To evaluate the performance of our nanocomposite heater, we performed a series of measurements. First, we applied constant current across standard test microelectrodes with identical geometries (see details in the Section 2) but with varying PVA concentrations and recorded their temperature after stabilization. As expected from the resistivity measurements (Figure 2e), the highest temperature (about 110 °C) is achieved with the 30% PVA composite (see Figure 3a). Assuming that the current contributes to Joule heating only, we can evaluate the consistency of our measurements by checking whether I^2^/ΔT is constant across the sets (R_ct_ = I^2^/ΔT). Taking standard deviations into account and room temperature of 27 °C, as recorded during the experiments (ΔT = T_meas_ − T_27_), our results are consistent within 10% for each set, except for the 0% PVA, which exhibits up to 20% differences. The differences are more notable for the low current (2A), where a slight temperature variation has a large influence on the results. 

We then performed measurements on the 30% PVA nanocomposite as a function of time (Figure 3b). It is interesting to note that it takes approximately 15 min to reach the plateau, independently of the current. Afterwards, the temperature is stable for the remainder of the measurement (SD < 3%).

Further experiments were performed to verify the heating performance of the nanocomposites after heating cycles (Figure 3c). A current of 7A was applied for 60 min to newly prepared samples. The samples were returned to room temperature in between each measurement cycle, as confirmed using an infrared (IR) camera. It can be seen that the temperature of the nanocomposite heater increases with each cycle, but it stabilizes after five cycles. To understand where this intriguing phenomenon comes from, we proceeded to a systematic study on the nanocomposite in its unprocessed (liquid) and processed form, before and after heating cycles.

### 3.3. Study of the Raw Materials Properties 

To evaluate the effect of the PVA concentration on the bulk properties of the nano-composite ink, we first performed viscosity measurements. This was carried out under general linear viscoelastic region (LVR) using Discovery Hybrid Rheometer HR 10 from TA Instrument (New Castle, DE, USA) with a gap dimension set to 0.328 mm and an upper plate of 7.935 mm radius. For the measurements, 1 mL of the nano-composite ink was used with shear rates between 20 s^−1^ and 200 s^−1^.

Figure 4a shows that pristine silver ink has a higher viscosity coefficient than any of the composite nano-inks. The overall decrease in the viscosity of the composites can be explained by the fact that PVA is widely used as a surfactant/dispersant in the synthesis of silver nanoparticles and the water added to dissolve the PVA. The figure also shows that the viscosity value is not significantly affected by the PVA concentration (here, 10%, 20% and 30% *w*/*w*). We also note that the rheological curves show hysteresis, i.e., thixotropic behavior, confirming the results reported elsewhere [40].

The thermal stability of the nanocomposite ink is evaluated using thermogravimetric analysis (TGA) to determine the fraction of volatile components by measuring the weight change during heating. TGA thus provides information on the thermal behavior of materials, including thermal decomposition and pyrolysis. The measurements were carried out using a Perkin Elmer DTA 4000 instrument (Waltham, MA, USA). Briefly, 15 mg of sample is placed in an aluminum pan and heated from room temperature to 600 °C at a rate of 20 °C/min under nitrogen atmosphere (30 mL/min). Figure 4b shows the TGA profiles of the nanocomposites in comparison to the pristine silver ink and PVA. In the case of pristine PVA (100%), the mass loss observed in temperature range of 87–120 °C is due to absorbed moisture [40]. Subsequently, a considerable mass loss occurs in the temperature range from 300 °C to 400 °C, attributed to the release of hydroxyl groups as water, aldehydes, and methyl ketones [40]. The third-stage of decomposition occurs above 400 °C, producing carbon and hydrocarbons as byproducts. 

The nanocomposites with ascending concentration of PVA generally follow the degradation curve of the pristine silver ink (PVA 0%) with two regions of interest. In the initial region, a sharp reduction in weight loss of around 200 °C is ascribed to the evaporation of the volatile compound of the silver ink, including the PVA solution. As the content rises, the residual weight of each nanocomposite aligns with the proportion of the PVA composition, with the 30% PVA composite showing the highest weight loss. However, the higher degradation rate of 30% PVA composite cannot be solely attributed to the excess PVA material, as evidenced by the lower weight loss of pure PVA (100%) in comparison to other composites. Notably, at 30% PVA, an earlier decomposition temperature at around 70 °C is found, which strongly suggests water evaporation from the distilled water used to dissolve the PVA powder. This observation is consistent with the viscosity of the 30% PVA being the lowest among the various silver ink composites (Figure 4a). The second region of weight loss manifests in a broad temperature range of 250 °C to 500 °C, indicating that breakdown products evolve into carbon and hydrocarbons, similar to the behavior found in pristine PVA. 

To complement the TGA measurements, we performed differential scanning calorimetry (DSC) experiments. The measurements were performed on a Perkin Elmer DTA 4000 equipment under a nitrogen atmosphere (30 mL/min). Figure 4c shows the curves for all samples. Figure 4d recapitulates the values automatically generated by the software. They include the glass transition temperature T_g_, the melting temperature T_m_, the melting enthalpy ΔH_m_, and the temperature of the onset of degradation T_onset_. The glass transition temperature (Tg) for pure silver nanoparticle paste (0% PVA) is 78.56 °C. With the addition of PVA at concentrations of 10%, 20%, and 30%, the Tg values gradually shift to higher temperatures, namely 83.28 °C, 89.58 °C, and 99.56 °C. The observed increase in Tg values indicates a reduction in polymer chain mobility due to the formation of hydrogen bonds between the hydroxyl groups (-OH) of PVA molecules and water. Furthermore, the interaction between Ag+ ions with PVA molecules may contribute to this process. Notably, Huang et al. [41] postulated that Ag+ ions can form a chelate with two hydroxyl groups of PVA, potentially constraining the mobility of PVA chains and, therefore, raising both Tg and melting temperature. Interestingly, the melting temperature of the pure silver nanoparticle paste (0% PVA) and the pristine PVA (100% PVA) are very close with 192.2 and 192.0 °C, respectively, but these values increase for the nanocomposite. It reaches a maximum temperature of 206.2 °C for the 30% PVA sample. This is consistent with other findings that indicate increased interaction between silver nanoparticles and the hydroxyl groups of the PVA [38].

When comparing both techniques, the TGA results show an increased weight loss with PVA concentration, whereas DSC shows an increased stability with PVA concentration. This may seem contradictory at first. But the TGA results may be explained by the presence of soluble compounds in the silver ink. Indeed, such a compound could be dissolved when mixing the aqueous PVA solution with the silver ink. There are a number of volatile compounds, including rosin, used in printing inks or solder reflux [42]. This behavior would also partly explain the lower viscosity noted for composites with higher PVA concentration (see Figure 4a). This hypothesis will be tested and discussed further in Section 3.4 below. 

### 3.4. Morphology and Composition of Processed Samples

Having characterized the unprocessed nanocomposite, we now evaluate processed electrodes (30% PVA) before and after the heating cycles. We first investigate the electrodes as deposited after doctor-blade processing. We used scanning electron microscopy to evaluate the morphology of the nanocomposites and the effect of the applied current (0–7 A) after 60 min. Figure 5a shows significant differences in the silver microstructures for each of the currents. The micrographs seem to indicate that the nanocomposite becomes more compact with current. To verify this observation, we applied the processing algorithm [34] introduced previously. Figure 5b shows the processed images. The apparent porosity (Figure 5c) clearly decreases with the current, with a porosity value above 40% for the 0 A condition and less than 20% for the sample subjected to 7 A. Using ANOVA (F = 42.81, *p* = 0.0001) and Tukey’s test, we note that all samples show a significant difference. 

Figure 5d shows that the Vicker’s hardness increases with the current. Using ANOVA (F = 16.75, *p* = 0.001) and Tukey’s test, we note that all samples show a significant difference, except between 0 A and 1 A. The hardness of the pristine sample (0A) is about 5.5 HV, while the sample subjected to 7 A reaches 7.8 HV, corresponding to a 42% improvement.

To verify whether the observed change in porosity is due to the heat or to heat combined with electric current (Joule heating), we applied the porosity measurement algorithm after heating samples on a hot plate (i.e., without electric current). As for the samples heated by Joule heating, the porosity of the hot-plate-heated sample decreases with the temperature. However, the absolute values are different, starting at about 60% for the room temperature condition and going down to 33% at 112 °C. In theory, both room-temperature conditions should give the same porosity (since no current or heat was applied to the samples) at this stage. The ~15% difference is attributed to the fact that each set of samples was prepared and measured on different days. Indeed, processing variability (for example, drying time, due to variations in ambient temperature and humidity or SEM stability) could induce morphology changes or measurement artifacts. To account for these, we have truncated 15% from the hot-plate results (so that both starting porosity values are around 45%). By doing so, the porosity measurement for each condition matches within 12%, which suggests that the porosity change observed is due to heating. 

We hypothesized above that this weight loss was in part due to volatile compounds evaporating after the heating cycles. The voids left by this “lost phase” could indeed lead to the re-organization of the silver nanoparticles. This re-organization is further promoted by the strong interaction between the hydroxyl group of the PVA and silver [38]. To further confirm this hypothesis, we have performed energy-dispersive X-ray spectroscopy (EDX) to compare the normalized composition of a heating pad before and after application of 1 V for 30 min (Table 1). Appendix A shows the spectra. The results of normalized weight concentration show that the relative silver level increases significantly with the application of current, while both carbon and oxygen levels decrease. Traces of gold were also noted and only changed marginally after heating. This observation confirms the loss of organic volatile compounds (C, O) due to heating, as depicted in Figure 6. 

To understand the effect of the cycles on heater performance, we performed further characterization on 30% PVA electrodes after heating cycles. A cycle consists of applying 7 A to the heaters for 60 min and then letting it cool down to room temperature before the next cycle. Figure 7a shows the SEM observation for the first three heating cycles. To extract meaningful information from the micrographs, we applied an image-processing algorithm [34] to calculate the apparent porosity of the composites. Using ANOVA (F = 3.97, *p* = 0.058) and Tukey’s test, there is no statistically significant difference between the cycles, except between the first and the third one. Microhardness measurement was also performed to evaluate the effect of the cycles. Figure 7c shows the values obtained, and ANOVA (F = 1.77, *p* = 0.226) with Tukey’s test indicate that there is no significant difference in hardness between the different cycles. The same applies to the resistivity (Figure 7d) that does not show any significant difference using ANOVA (F = 2.42, *p* = 0.170) and Tukey’s test. The above test concludes that the main improvements come directly after the first heating cycles. 

### 3.5. Validation Using DNA Amplification

Having confirmed our hypothesis that links performance improvement with the loss of volatile compounds, we evaluate the performance of our nanocomposite heater against a gold standard. We performed a LAMP test to amplify and detect DNA from Trypanosoma brucei, a parasite associated with African sleeping sickness. LAMP tests are based on isothermal amplification of target DNA [43] and require stable heating for up to 1 h. 

We applied our nanocomposite (30% PVA) heating electrodes on the backside of a commercially available chip, using a doctor blade as described above, and connected it to a DC power supply (See Figure 8). The electrodes were subjected to three heating cycles to reach the optimal performance before the amplification step, as explained above. After introducing the samples and reagents into the chip, we performed the isothermal amplification at 60 °C by applying 4.5 A for 60 min. After this step, we retrieved the amplified product and used gel electrophoresis to verify the length of the amplified DNA segments. For comparison, we performed the same procedure on a commercial system. Our results show that our solution compares well with its commercial counterpart as a band around 100 bp, corresponding to the expected 103 bp amplicon, is visible in both cases (Figure 8c). Negative and positive controls confirm the validity of our experimental approach. 

## 4. Conclusions

In this manuscript, we have demonstrated the development, characterization, and validation of a high-performance nanocomposite (PVA–silver ink) heater deposited using the doctor-blade technique. We studied the composite in its unprocessed and processed form, including after heating cycles. We show that the electrical and mechanical performance of the heater improved by 35% and 42%, respectively, after the first heating cycles. This surprising improvement is attributed to the removal of performance hindering organic compounds present in the silver ink. The compound dissolves in the aqueous PVA solution and evaporates upon heating as confirmed by EDX measurements. This removal leads to the reorganization of the nanocomposite and favors a stronger interaction between the silver and PVA [38], thereby increasing both the resistivity and the hardness of the electrodes. To validate our nanocomposite heater, we deposited it onto a microfluidic chip to perform isothermal amplification. We successfully amplified and detected DNA from Trypanosoma brucei, associated with African sleeping sickness, thereby confirming the applicability of our microheater for healthcare applications. Overall, our results show that the silver ink–PVA composite can greatly enhance the performance of heaters, and we anticipate that it will be used in a wide range of applications since it relies on well-established fabrication methods without additional constraints.

## Figures and Tables

**Figure 1 polymers-16-01164-f001:**
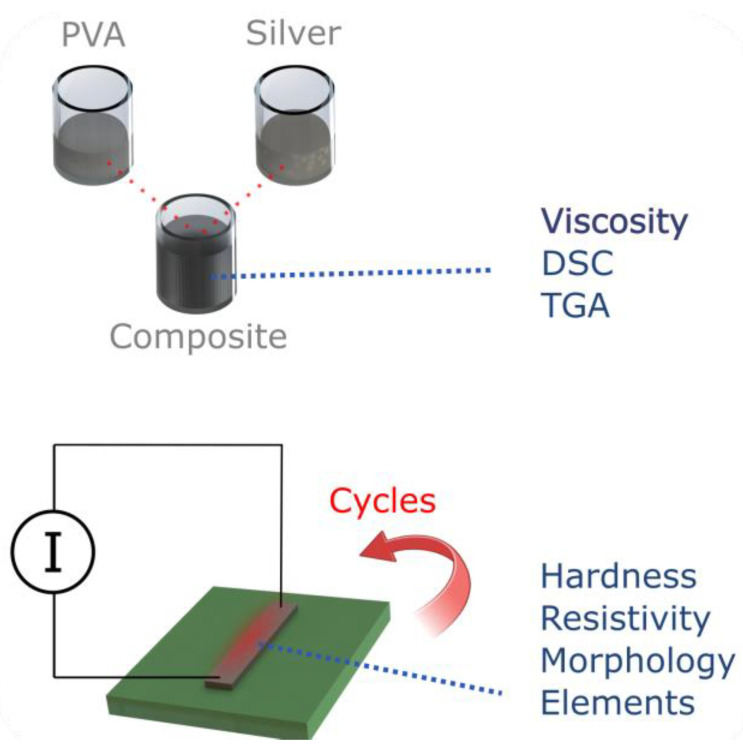
Overview of the project and materials characterization. Tests were performed on the bulk, unprocessed material and on the processed (structured) material before and after heating, including after a several cycles.

**Figure 2 polymers-16-01164-f002:**
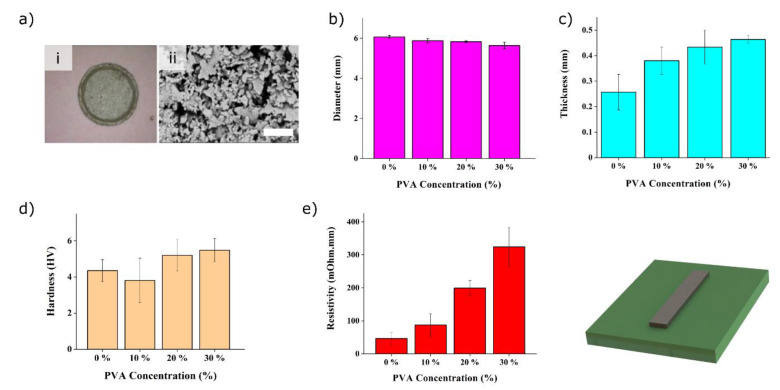
Processed samples characterization. (**a**) Optical image of the circular test pattern obtained with a silver 30% PVA nanocomposite (i) and scanning electron micrograph detail thereof (ii). The scale bar is 5 µm. (**b**) Measurement of diameter and (**c**) thickness of the printed silver electrode specimen with various concentrations of PVA. (**d**) Vickers hardness of the electrode as a function of the PVA concentration. (**e**) Resistivity of the nanocomposites of varying PVA concentrations from pristine silver ink (0% PVA) to 30% PVA nanocomposites.

**Figure 3 polymers-16-01164-f003:**
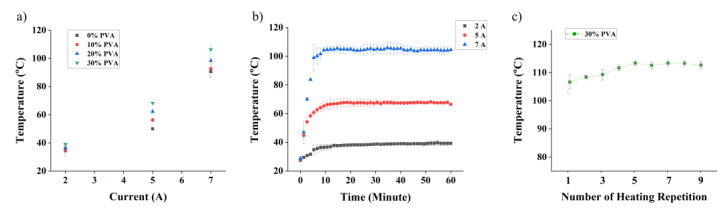
Nanocomposite heater performance. (**a**) Measurement of temperature of nanocomposite electrodes with concentrations of PVA ranging from 0 to 30% as function of current. (**b**) Evolution of the temperature of 30% PVA nanocomposite heater as a function of time for different currents. (**c**) Repetitive heating cycles are applied on a 30% PVA sample (constant current of 7A). The temperature increases for five cycles and then stabilizes.

**Figure 4 polymers-16-01164-f004:**
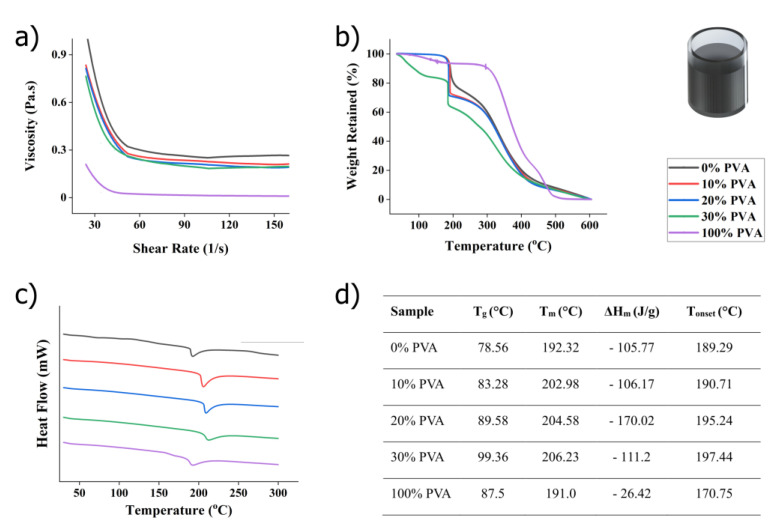
Unprocessed samples characterization. (**a**) Graph showing the viscosity of silver/PLA composite. Pristine silver ink (0% PVA) shows higher viscosity than samples with PVA (10%, 20%, and 30%). (**b**) Thermogravimetric analysis graph showing silver/PLA composite at various concentrations. (**c**) Result of differential scanning calorimetry (DSC) of the silver/PVA electrode with various PVA concentration. (**d**) Data extracted from the DSC analysis, where T_g_ is the glass transition temperature; T_m_ the melting temperature; ΔH_m_ the materials melting enthalpy; and T_onset_ the temperature of the onset of degradation. All data were repeated in triplicates and exhibited consistent results.

**Figure 5 polymers-16-01164-f005:**
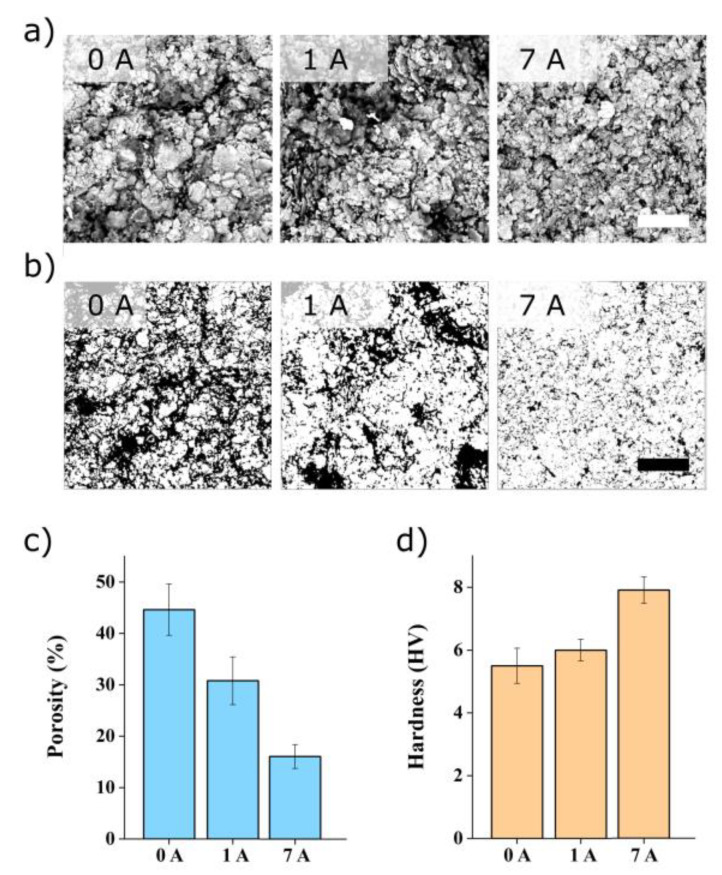
Evaluation of the effect of current on 30% PVA electrodes using (**a**) scanning electron micrograph of the nanocomposites after 60 min with a current applied between 0 and 7 A and (**b**) visualization of porosity after image processing (**c**) graph showing the quantified porosity based on the images, with 20 µm scale bars. (**d**) Hardness of the 30% PVA nanocomposite as a function of the applied current.

**Figure 6 polymers-16-01164-f006:**
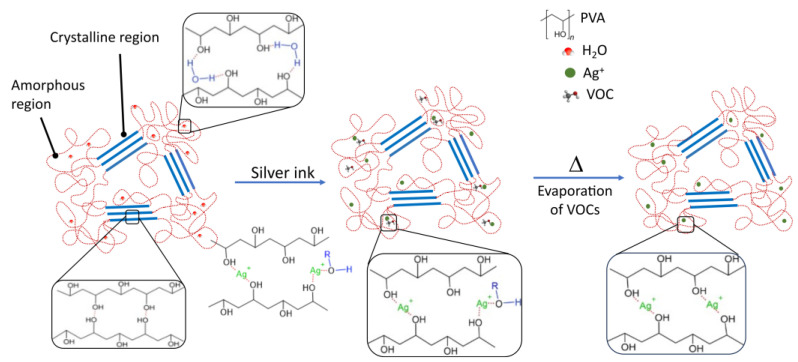
Illustration of the intermolecular interactions between the silver ink and PVA and the effect of heating cycles, during which a volatile organic compound evaporates, promoting the interaction between the silver ions Ag^+^ and PVA.

**Figure 7 polymers-16-01164-f007:**
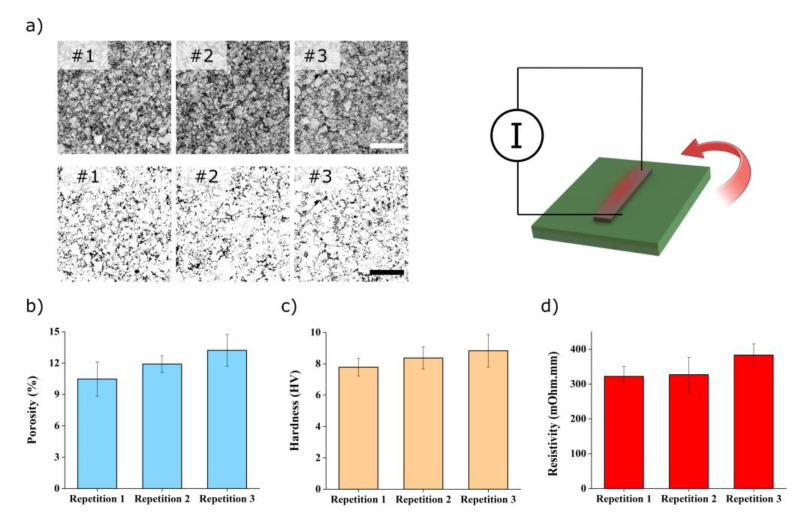
Sample characterization after heating cycle of 30% PVA nanocomposite electrode. (**a**) Scanning electron micrographs showing the evolution of the samples’ morphology after heating cycles (#1–#3) and corresponding porosity after image processing. The scale bars are 20 µm. Evolution of (**b**) the porosity, (**c**) the hardness, and (**d**) the resistivity after heating cycles.

**Figure 8 polymers-16-01164-f008:**
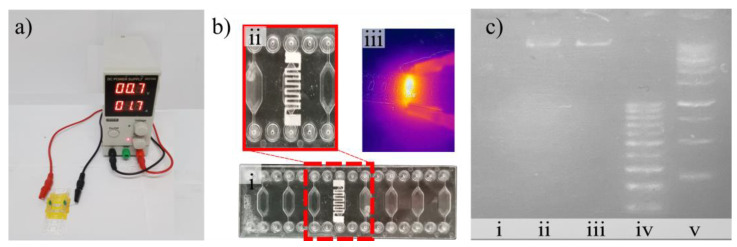
Validation of microheater for isothermal DNA amplification using LAMP procedure. The image in (**a**) shows the set-up used to heat the chip, (**b**) provides details of the electrodes on the chip (i,ii) as well as the measure of its heating using infrared camera (iii), and (**c**) shows the results from the LAMP process, where i and ii corresponds to the negative and positive controls on our chip, iii shows the positive control on a commercial system, and iv and v represent the ladders.

**Table 1 polymers-16-01164-t001:** Energy-dispersive X-ray spectroscopy data show that the relative silver content (normalized weight concentration) increases while the carbon and oxygen concentration decrease after 7 A has been applied to the electrode.

	0 A	7 A
Silver	52.84%	74.38%
Carbon	25.16%	11.54%
Oxygen	17.69%	8.58%
Gold	4.31%	5.50%

## Data Availability

Raw data are available at https://zenodo.org/records/8152624 (accessed on 10 April 2024).

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
