# Peer review of "High-Performance, Easy-to-Fabricate, Nanocomposite Heater for Life Sciences and Biomedical Applications"

_polymers, 2024, doi:10.3390/polym16081164_

Round 1

Reviewer 1 Report

Comments and Suggestions for Authors

I have some main concerns about this paper:

1. Physical and important issue: I do not understand why, at constant applied voltage, an increase of resistivity leads to an increase of power dissipation. It should be the opposite.

2. I have not found the supplementary Materials

3. Fig. 5 is missing.

Author Response

We would like to thank the reviewer for taking the time to evaluate our manuscript and for giving us the opportunity to revise it. We have addressed each of the comments below, and the manuscript accordingly.

  1. Physical and important issue: I do not understand why, at constant applied voltage, an increase of resistivity leads to an increase of power dissipation. It should be the opposite.

We agree that our manuscript can be misleading as we do indeed apply a constant voltage but refer to an equation (1) that relates “current to heating power”.

The reason behind this is that the Joule heating (also called resistive heating) phenomenon at the microscale is linked to kinetic energy loss by charges (electrons in our case) as they travel through the conductor. This is why Joule heating is primarily linked to the formula P=RI2. Using this equation, it is obvious that the heating power increases with resistance.

However, as noted also in the manuscript, the power can be written as P=U2/R. This notation could indeed indicate that the power decreases when the resistance increases. However, this notation comes from Ohm’s law (U=RI).

P  =RI2

   =R(U2/R2)

   =U2/R

And it is also the reason why the power decreases. Indeed, when applying Ohm’s law we see that the current decreases with the resistance, for a constant voltage (I=U/R). In other words, there will be less current if the resistance is large. This is why the power decreases with increasing resistance, but fundamentally and physically, the heating power is proportional to the resistance, since it is linked to the losses of charges travelling through a conductor (the current).

However as noted above, we agree that our manuscript can be misleading. This is why we have updated it in the introduction (lines 45-46) to clarify the situation.

  1. I have not found the supplementary Materials

We are sorry that the reviewer could not find the supplementary information, but we double checked, and it was uploaded correctly in the system. We do not know why it was not made available. We have informed the editor about it. Hopefully it will be resolved soon.

  1. 5 is missing.

We thank the reviewer for pointing out this issue. This was due to poor labelling in the manuscript. We have now updated it and each figure is referred to correctly.

Reviewer 2 Report

Comments and Suggestions for Authors

The manuscript "High performance, easy to fabricate, composite microheater for life sciences and biomedical applications" focuses on an interesting field in microtechnology that is design, fabrication and testing of novel microheaters.

Such devices are particularly needed for various technological and biomedical microfluidic applications where precise temperature control is a key prerequisite for successful procedures of chemical analysis or synthesis.

In this respect, the manuscript topic is interesting and urgent.

A thorough manuscript analysis reveals, however, certain flaws and imperfections as well as journal suitability concerns.

1. The manuscript is mostly technical and focuses on device design and fabrication from PVA and silver ink. Although the authors introduce this material as a "nanocomposite" the research focus in not on polymer and it impact on the properties of material and performance of a microheater. In a manuscript composed specifically for Polymers, it is expected to predominantly focus on the experiments and discussion on polymer impact on the structure and performance of the composite, macromolecular aspects of silver-polymer interactions in the composite, or a structure-property relationship, similarly to what was provided in Lines 345-370.

It is doubtful that the manuscript fits the scope of Polymers. It seems to be more relevant for another journal, which focuses in microdevices and microtechnology.

2. PVA is a too well-studied and thus simplistic research object as a polymer. Why did authors select this polymer material for producing a novel microheater among other alternatives?

3. In Fig. 3, the reported diameter of a microheater is several mm and thickness reaches half a millimeter. Such a size range is far above microfluidic or (generally) microscale range of sizes although still suitable for certain large-size microchannels. Is it possible to produce smaller microheaters with the diameter about 100-200 µm? Otherwise, the term "microheater" is not suitable for such a device.

4. Paragraph 3.5 is misleading and requires a thorough revision for a better readability. At its current state, it does not clarify how DNA amplification results relate to the validity of the produced microheater.

5. The abstract does not fit the requirements of Polymers and should be revised according to the manuscript preparation guide.

6. The literature review should start in the Introduction, not in the Abstract.

7. The authors should thoroughly check formatting of the manuscript text and citations as well as English quality.

8. At its best, the manuscript represents some promising technical results and is recommended to be submitted to a technical journal as the paper seems to be unsuitable to Polymers. Alternatively, the manuscript can be seriously revised with the focus on polymer behavior in the nanocomposite during operation cycles and (recommended) the related structure-performance relationship.

Comments on the Quality of English Language

English language requires moderate editing. The manuscript text is easy to understand. The text is, however, recommended to be revised by a native speaker for smooth reading and stylistic consistency.

Author Response

We thank the reviewer for taking the time to evaluate our manuscript and for giving us the opportunity to revise it. We have addressed each of your comments and revised the manuscript accordingly. We hope that the reviewer will find the new version of the manuscript worthy of a publication in Polymers.

  1. The manuscript is mostly technical and focuses on device design and fabrication from PVA and silver ink. Although the authors introduce this material as a "nanocomposite" the research focus in not on polymer and it impact on the properties of material and performance of a microheater. In a manuscript composed specifically for Polymers, it is expected to predominantly focus on the experiments and discussion on polymer impact on the structure and performance of the composite, macromolecular aspects of silver-polymer interactions in the composite, or a structure-property relationship, similarly to what was provided in Lines 345-370.

We thank the reviewer for pointing out an apparent mismatch between the content of the manuscript and the focus of the journal. We take full responsibility for this misunderstanding, but we believe that rather than being a mismatch, the issue is due to a poor communication from us.

Indeed, the main finding of our manuscript is that the addition of PVA (in aqueous solution) to silver ink results in the solubilization of a performance hindering organic compound. This compound then evaporates during the heating process thus improving the performance of the heater. All the experiments done on the bulk, processed and cycled composited shed new evidence for this finding. However, we agree that this finding was poorly conveyed in the original manuscript. This is why we have thoroughly revised it to emphasise this finding, which links structure/composition with performance. In particular, we have re-ordered the sections, and split figures to show more clearly our investigation that relate structure to property. We have also emphasized the discussion on the DSC and TGA measurements.

  1. PVA is a too well-studied and thus simplistic research object as a polymer. Why did authors select this polymer material for producing a novel microheater among other alternatives?

We agree with the reviewer that PVA is a well-studied material. It was chosen primarily due to its availability, including in Low- and middle-income countries such as Indonesia where the majority of the work was performed. It was also chosen for its low toxicity and dissolvability. We have now added a sentence to justify our choice in the Introduction (lines 81-84).

  1. In Fig. 3, the reported diameter of a microheater is several mm and thickness reaches half a millimeter. Such a size range is far above microfluidic or (generally) microscale range of sizes although still suitable for certain large-size microchannels. Is it possible to produce smaller microheaters with the diameter about 100-200 µm? Otherwise, the term "microheater" is not suitable for such a device.

We agree with the reviewer that the term microheater is not appropriate given the type of structures proposed in the manuscript. Even though the techniques used in the manuscript can be used to create microstructures, it is not the case in our manuscript. Therefore, we have removed the term “micro”heater in favour of composite heater.

  1. Paragraph 3.5 is misleading and requires a thorough revision for a better readability. At its current state, it does not clarify how DNA amplification results relate to the validity of the produced microheater.

We thank the reviewer for pointing this out and we agree that the original manuscript did not convey our case very strongly. We have therefore updated the appropriate section (3.5) to clarify the situation.

  1. The abstract does not fit the requirements of Polymers and should be revised according to the manuscript preparation guide.

We thank you for pointing this out. We updated the abstract and removed the citations from the abstract to comply to the editorial policy of Polymers.

  1. The literature review should start in the Introduction, not in the Abstract.

See point 5 above.

  1. The authors should thoroughly check formatting of the manuscript text and citations as well as English quality.

We thank the reviewer for the suggestions. We have now thoroughly reviewed the manuscript and have updated many parts to improve its readability.

  1. At its best, the manuscript represents some promising technical results and is recommended to be submitted to a technical journal as the paper seems to be unsuitable to Polymers. Alternatively, the manuscript can be seriously revised with the focus on polymer behavior in the nanocomposite during operation cycles and (recommended) the related structure-performance relationship.

We hope that the changes brought to the manuscript now reflect better the finding we would like to report.

Reviewer 3 Report

Comments and Suggestions for Authors

Microheaters have extensive application value in biomedical fields.

1. The parameter symbols for the conductivity coefficient in Equation 1 should be clear (two different ones appear in the text).

2. Is there any repetition in the experiment in Figure 2, and there are also some data with different significant digits in d, why?

3. Figures 5 and 6 are incorrectly labeled as Figures 6 and 7.

Author Response

We would like to thank the reviewer for their overall positive assessment of our manuscript and for taking the time to evaluate it. We have addressed each of the comments below, and the manuscript accordingly.

  1. The parameter symbols for the conductivity coefficient in Equation 1 should be clear (two different ones appear in the text).

We thank the reviewer for pointing out this inconsistency. We have now corrected the manuscript accordingly.

  1. Is there any repetition in the experiment in Figure 2, and there are also some data with different significant digits in d, why?

We thank the reviewer for their important questions. The experiments were repeated on at least 3 independent occasions and the results were consistent each time. We have now updated the figure caption to add this important information. Regarding the significant digits, it is only due to poor rounding error (e.g., 87.5 should read 87.50). We have now updated the values.

  1. Figures 5 and 6 are incorrectly labeled as Figures 6 and 7.

We thank the reviewer for pointing out this issue. We have now corrected the manuscript accordingly.

Round 2

Reviewer 1 Report

Comments and Suggestions for Authors

Author's explanation about the increase of power heating with increasing resistance, at constant voltage,  is still not clear to me.

Which are the currents that you impose in the four-point method to get 1V, over the inner electrodes, at the different PVA concentrations?

Author Response

We wouldn’t like to thank the reviewer for their comments and the time they took to help us improve our manuscript. You will find below the answers to their questions.

  • Author's explanation about the increase of power heating with increasing resistance, at constant voltage,  is still not clear to me.

In order to clarify the situation, we propose to conceptualise a simple heating resistance (represented by ----WWW---- ) as indicated below

The right-hand terminal is connected to the ground (represented by //). In the case of an imposed current on the left-hand terminal, we have “conventional” Joule Heating, with the dissipated power given by P=RI2. In this case, if R increases, then the power will increase.

(I)  ----WWW----// 

In case the voltage is imposed, we have the following situation

(U)  ----WWW----// 

Using Ohm’s law, we note that the current I =U/R decreases with increasing resistance. It is this reduction in current that reduces the heating power (U=I1/R1 = I2/R2, if R2 =2R1, then I2 = I1/2 to keep the voltage U constant). This approach does not violate Joule’s law that show that heating is due to the collision of moving charge carriers in a resistive medium.

To provide more details for the interested reader about this topic, but without distracting them from the main message of the paper, we have added a reference (line 47).

  • Which are the currents that you impose in the four-point method to get 1V, over the inner electrodes, at the different PVA concentrations?

We thank the reviewer for this question. The four-point probe method (also called the Kelvin method) consists of four equally spaced, co-linear electrical probes. Current is applied between the two outer electrodes and the voltage is measured between the two middle ones. This configuration eliminates contact and wire resistance. The four-point probe methods was used in our study to measure the electrodes’ resistivities at all PVA concentrations.

We have captured this information in the supplementary information where the method is shown so as not to distract the reader from the main message of the manuscript.

Reviewer 2 Report

Comments and Suggestions for Authors

The authors performed a considerable improvement of the manuscript.

The focus on a PVA-based nanocomposite and its properties and impact on heating performance makes the paper more suitable for the scope of Polymers and the respective Special Issue.

The authors addressed other comments and concerns such as DNA amplification method details.

The manuscript can be considered for publication in Polymers. However, the following issues should be addressed:

1. Some of the revision text (blue color) in Lines 78-123 seems to be more relevant for Results and Discussion. Introduction should include a problem statement with appropriate references, the aim of research, and main research findings.

2. Line 418-420. Can evaporation of volatile compounds affect performance of a device, which incorporates a composite heater, such as a microfluidic chip? Did the authors perform such an evaluation for DNA amplification tests? Provide an appropriate discussion.

3. The authors are recommended to introduce a figure (model) of a suggested PVA nanocomposite structure on a molecular level, which can also demonstrate the main effect discussed in this manuscript (evaporation of volatile compounds).

4. Lines 452-456 - a proper reference should be provided.

5. Conclusions must be revised according to the focus of the revised manuscript. At the current state, they remain unmodified (including terminology in Line 477).

Comments on the Quality of English Language

Minor language editing is recommended.

Author Response

We are truly grateful to the reviewer for the time they took to review our manuscript and importantly for their valuable comments that will help us further improve it. We also thank them for the positive comments on our modifications after the first round of revision. You will find below the answers to the remaining issues that were pointed out.

  1. Some of the revision text (blue color) in Lines 78-123 seems to be more relevant for Results and Discussion. Introduction should include a problem statement with appropriate references, the aim of research, and main research findings.

We thank the reviewer for pointing this out. We agree that the sections highlighted needed an improvement. We have removed superfluous information and simplified the text wherever possible. We currently have a structure that follows the reviewer’s recommendations but rather than it being linear as suggested, it is implemented into a narrative that highlights the progression of the work (rationale) and the sections. The current version, which is in line with editorial guidelines, emphasises the relevance of the work, as suggested by the reviewer during the first round of revision.

  1. Line 418-420. Can evaporation of volatile compounds affect performance of a device, which incorporates a composite heater, such as a microfluidic chip? Did the authors perform such an evaluation for DNA amplification tests? Provide an appropriate discussion.

We thank the reviewer for this insightful comment. Indeed, we cannot exclude that the volatile compound will not interact with the assay. However, it is very unlikely as the electrodes are positioned outside of the channel (on the chip directly) and hence they are never in direct contact with the liquids/reagents. Moreover, we have performed the heating cycle three times before the amplification experiment (see lines 459). This is to improve the performance of the electrodes, but it will also ensure that the volatile compound has evaporated before the DNA amplification step. Finally, the negative control did not reveal any substantial changes. Nevertheless, we agree that the fact that the electrodes are on the outside of the channel is not made clear enough and could lead to confusion. We have therefore updated the manuscript accordingly (lines 457).

  1. The authors are recommended to introduce a figure (model) of a suggested PVA nanocomposite structure on a molecular level, which can also demonstrate the main effect discussed in this manuscript (evaporation of volatile compounds).

We thank the reviewer for this excellent suggestion. We have now added a figure (Fig. 6) and even integrated part of it into the Graphical TOC.

  1. Lines 452-456 - a proper reference should be provided.

We thank the reviewer for this suggestion. We have now added a reference as suggested.

  1. Conclusions must be revised according to the focus of the revised manuscript. At the current state, they remain unmodified (including terminology in Line 477).

We thank the reviewer for pointing out this oversight. We have now updated the conclusions to be more in focus with the new version of the manuscript.

Round 3

Reviewer 1 Report

Comments and Suggestions for Authors

I still do not understand why at fixed voltage the temperature increases with increasing resistivity. Indeed at fixed voltage, current decreases with increasing resistivity and therefore dissipated power decreases.

Author Response

Dear reviewer, 

we are really sorry that we failed to understand your point and have missed this issue previously. Indeed, as we explained in the manuscript and in our previous responses the temperature should decrease at fixed voltage with increasing resistivity. However, our results suggest otherwise (in particular Fig. 3).

Thankfully, our results do not violate the laws of physics (more on that below) and the mistake was not intentional. Indeed, after looking through the picture taken by the student we realised that even though he had dialed the voltages as per the results reported in the manuscript, the DC power supply was set to current mode as shown on sample pictures below.

However, we will now need to redo the experiments reported in Fig. 3 to validate that the current is constant for each of the conditions tested (as suggested by the error bars on Fig. 3a). Then we will have to correct the rest of the manuscript accordingly (e.g., reporting the results as function of current rather than voltage and update the methods and discussions).

We will ask the editors for grant us additional time (20 days) to address the issues.

Finally we are grateful to the reviewer for their feedback that will help us produce a much better manuscript.

Reviewer 2 Report

Comments and Suggestions for Authors

The authors considered my recommendations and concerns. The manuscript can be a good contribution to the field.

Author Response

We thank the reviewer for their comment and their support that has contributed to strengthen our manuscript.